# Alpha-Glucosidase Inhibitory Diterpenes from *Euphorbia antiquorum* Growing in Vietnam

**DOI:** 10.3390/molecules26082257

**Published:** 2021-04-13

**Authors:** Cong-Luan Tran, Thi-Bich-Ngoc Dao, Thanh-Nha Tran, Dinh-Tri Mai, Thi-Minh-Dinh Tran, Nguyen-Minh-An Tran, Van-Son Dang, Thi-Xuyen Vo, Thuc-Huy Duong, Jirapast Sichaem

**Affiliations:** 1Faculty of Pharmacy and Nursery, Tay Do University, Can Tho 94000, Vietnam; tcluan@tdu.edu.vn; 2Department of Chemistry, University of Education, 280 An Duong Vuong Street, District 5, Ho Chi Minh City 72711, Vietnam; ngocdaosph@gmail.com (T.-B.-N.D.); thanhnhaksb@gmail.com (T.-N.T.); 3Vietnam Academy of Science and Technology, Graduate University of Science and Technology, 18 Hoang Quoc Viet, Cau Giay, Ha Noi 11355, Vietnam; maidinhtri@gmail.com (D.-T.M.); dvsonitb@gmail.com (V.-S.D.); 4Institute of Chemical Technology, Vietnam Academy of Science and Technology, 01 Mac Dinh Chi, Ho Chi Minh City 71007, Vietnam; 5Department of Biology, University of Education, 280 An Duong Vuong Street, District 5, Ho Chi Minh City 72711, Vietnam; dinhttm@hcmue.edu.vn; 6Faculty of Chemical Engineering, Industrial University of Ho Chi Minh City, Ho Chi Minh City 71420, Vietnam; trannguyenminhan@iuh.edu.vn; 7Vietnam Academy of Science and Technology, Institute of Tropical Biology, Ho Chi Minh City 71308, Vietnam; 8Faculty of Technology, Van Lang University, Ho Chi Minh City 71013, Vietnam; xuyen.vt@vlu.edu.vn; 9Research Unit in Natural Products Chemistry and Bioactivities, Faculty of Science and Technology, Thammasat University Lampang Center, Lampang 52190, Thailand

**Keywords:** Euphorbiaceae, *Euphorbia antiquorum* L., *ent*-atisane, diterpenoid, alpha-glucosidase inhibition

## Abstract

Bioactive-guided phytochemical investigation of *Euphorbia antiquorum* L. growing in Vietnam led to the isolation of five *ent*-atisanes, one *seco*-*ent*-atisane, and one lathyrane (ingol-type). The structures were elucidated as *ent*-1*α,*3*α,*16*β,*17-tetrahydroxyatisane (**1**), ethyl *ent*-3,4-*seco*-4,16*β*,17-trihydroxyatisane-3-carboxylate (**2**), *ent*-atisane-3-oxo-16*β*,17-acetonide (**3**), *ent*-3*α*-acetoxy-16*β*,17-dihydroxyatisane (**4**), *ent*-16*β*,17-dihydroxyatisane-3-one (**5**), calliterpenone (**6**), and ingol 12-acetate (**7**). Their chemical structures were unambiguously determined by analysis of one-dimensional (1D) and two-dimensional (2D) nuclear magnetic resonance (NMR) and high resolution mass spectrometry, as well as by comparison with literature data. Among them, **1** is a new compound while **2** is an ethylated artifact of *ent*-3,4-*seco*-4,16*β*,17-trihydroxyatisane-3-carboxylic acid, a new compound. Isolates were evaluated for alpha-glucosidase inhibition. Compound **3** showed the most significant inhibitory activity against alpha-glucosidase with an IC_50_ value of 69.62 µM. Further study on mechanism underlying yeast alpha-glucosidase inhibition indicated that **3** could retard the enzyme function by noncompetitive.

## 1. Introduction

*Euphorbia antiquorum* L., a medicinal plant, has traditionally been used for various treatments, for example, the latex has been used for emetic, purgative, and diuretic treatments, and the fresh stems for treatment of skin sores, scabies, and toothache [1,2,3]. Previous chemical studies on *E. antiquorum* in Thailand, Vietnam, and China have reported the wealth of diterpenoids, including lathyane, *ent*-atisane, *ent*-abietane, and *ent*-kaurane types [3,4,5,6,7,8]. Those isolated compounds showed various biological activities including cytotoxic, anti-inflammatory, anti-HIV activities, and the inhibition of nitric oxide (NO) production [9]. As a continuation of our discovery of alpha-glucosidase inhibitory metabolites from *E. antiquorum* [10], the further investigation on the Vietnamese plant *E. antiquorum* L. was performed. In this paper, the isolation and structural elucidation of seven diterpenoids, *ent*-1*α,*3*α,*16*β,*17-tetrahydroxyatisane (**1**), ethyl *ent*-3,4-*seco*-4,16*β*,17-trihydroxyatisane-3-carboxylate (**2**), *ent*-atisane-3-oxo-16*β*,17-acetonide (**3**) [11], *ent*-3*α*-acetoxy-16*β*,17-dihydroxyatisane (**4**) [12], *ent*-16*β*,17-dihydroxyatisane-3-one (**5**) [13], calliterpenone (**6**) [14], and ingol 12-acetate (**7**) [15] (Figure 1) from the most bioactive fraction of the title plant are reported. Their structures were elucidated by spectroscopic data analysis and compared with literature data. Isolates were assayed for their alpha-glucosidase inhibition.

## 2. Results and Discussion

### 2.1. Phytochemical Identification

Compound **1** was obtained as a colorless gum. Its molecular formula was deuced as C_20_H_34_O_4_ by the sodiated ion [M + Na]^+^ at *m*/*z* 361.2336 (calculated for 361.2355) in high resolution electrospray ionization mass spectroscopy (HRESIMS) spectrum. The ^1^H-NMR spectrum showed the three singlet methyl groups (*δ*_H_ 0.85, 0.97, and 1.00), two oxymethine protons (δ_H_ 3.41, t, ^3^*J*_H–H_ = 3.2 Hz and 3.52, t, ^3^*J*_H–H_ = 3.2 Hz), one oxymethylene proton (δ_H_ 3.34, d, ^2^*J*_H-H_ = 11.2 Hz and 3.48, d, ^2^*J*_H-H_ = 11.6 Hz). The ^13^C NMR spectrum, in conjunction with the heteronuclear single quantum coherence (HSQC) spectrum exhibited the resonances of 20 carbon atoms including five methine carbons (δ_C_ 33.1, 44.0, 44.5, 73.6, and 78.8, two latter oxygenated), eight methylene carbons (δ_C_ 19.4, 23.3, 24.5, 28.6, 30.3, 40.5, 54.1, and 69.7), three methyl carbons (δ_C_ 15.1, 22.3, and 29.1), and four quaternary carbons (δ_C_ 34.0, 38.8, 42.8, and 75.2). The above characteristic data suggested that **1** had an *ent*-atisane scaffold [3,10,16], further supported by the key heteronuclear multiple bond correlation (HMBC) correlations (Figure 2 and Appendix A). Detailed comparison of nuclear magnetic resonance (NMR) data of **1**, *ent*-1*β*,3*β*,16*β*,17-tetrahydroxyatisane and *ent*-1*β*,3*α*,16*β*,17-tetrahydroxyatisane [16] indicated that they shared the same planar structure. Indeed, the presence of the hydroxyl groups at C-1 and C-3 were defined by HMBC correlations of both H-18 (0.85, s) and H-19 (0.97, s) to C-3 (δ_C_ 78.8), C-4 (δ_C_ 38.8), and C-5 (δ_C_ 44.5) and of H-20 (1.00, s) to C-1 (δ_C_ 73.6), C-5, C-9 (δ_C_ 44.0), and C-10 (δ_C_ 42.8). The marked differences between **1** and previously mentioned compounds were the configurations of C-1 and C-3. Particularly, the equatorial orientation of H-1 and H-3 were deduced from their small coupling constants: *J*_H-1/H-2a_ 2.0 Hz and *J*_H-3/H-2a_ 3.2 Hz. This finding was further strengthened by nuclear overhauser effect spectroscopy (NOESY) correlations. Indeed, NOESY correlations of H-1/H-2a, H-3/H-2a, H-2a/H-20, H-20/H-13, and H-20/H-14 indicated the co-facial of all mentioned protons. The orientation of 16-OH was validated by NOESY correlations of H-17/H-9 and H-9/H-5 (Figure 3 and Appendix A), supported by the NMR comparison of the previously reported *ent*-atisanes, isolated from the same bio-source [3,10]. Altogether, the chemical structure of **1** was established as shown, namely *ent*-1*α,*3*α,*1*6β,*17-tetrahydroxyatisane.

Compound **2** might be an artifact of *ent*-3,4-*seco*-4,16*β*,17-trihydroxyatisane-3-carboxylic acid when using ethyl acetate during the extraction. It is worth noting that the mother compounds of **2** could be either methyl *ent*-3,4-*seco*-4,16*β*,17-trihydroxyatisane-3-carboxylate or *ent*-3,4-*seco*-4,16*β*,17-trihydroxyatisane-3-carboxylic acid which were new compounds. The occurrence of **2** proposed that *ent*-3,4-*seco*-4,16*β*,17-trihydroxyatisane-3-carboxylic acid or methyl *ent*-3,4-*seco*-4,16*β*,17-trihydroxyatisane-3-carboxylate were original compounds of the plant *E. antiquorum* growing in Vietnam. Up to now, very few *ent*-3,4-*seco*-atisanes have been reported in *E. antiquorum* [11]. Therefore, biologically active new chemical components from this plant might yet be found.

Compounds **1**–**4**, **6**, and **7** were evaluated for their alpha-glucosidase inhibition (Table 1). All tested compounds showed stronger activity than the positive control, acarbose (IC_50_ 332.5 µM), similar to previously reported *ent*-atisane [10]. Among them, compound **3** showed the highest alpha-glucosidase inhibition (IC_50_ 69.62 µM), indicating the important role of the acetonide moiety at C-16 and C-17. Compound **3** was prepared rapidly when **5** reacted with acetone under acidic catalyst at room temperature in one day. This indicated that **3** was an artifact of **5** when using acetone during the isolation.

### 2.2. Alpha-Glucosidase Inhibitory Activity of Isolated Compounds

The in vitro alpha-glucosidase inhibitory activity of **1**–**4**, **6**, and **7** was evaluated. All compounds displayed significant alpha-glucosidase inhibitory activity with IC_50_ values in the range of 69.62–156.14 µM. The inhibition of isolated compounds on other glycosidases should be evaluated to determine the selectivity. Unfortunately, these tests were not performed due to the minute amounts of isolated compounds.

### 2.3. Inhibition Type and Inhibition Constants of ***3*** on Alpha-Glucosidase

In order to examine the inhibition mechanism of **3**, their activity was measured at the different concentration of 4-nitrophenyl *β*-D-glucopyranoside (*p*NPG). The Lineweaver–Burk plots of a kinetic study of **3** (Figure 4A) showed linearity at each concentration examined (0, 13.8, 27.7, and 55.5 µM), which all intersected the x-axis in the second quadrant. The kinetic analysis revealed that *V*_max_ decreased while *K*_m_ remained constant, which showed that **3** acted as a noncompetitive inhibitor. The inhibition constant (*K*_i_) was 65.8 µM (Figure 4B).

## 3. Materials and Methods

### 3.1. Source of the Plant Material

The aerial parts of *E. antiquorum* were collected in Binh Thuan province, Vietnam. The scientific name of the plant was determined by Dr. Tran Cong Luan, Faculty of Pharmacy and Nursery, Tay Do University, Can Tho, Vietnam. A voucher specimen of *E. antiquorum* (No UP B007) was deposited in the herbarium of the Department of Organic Chemistry, Ho Chi Minh City University of Science, National University—HCMC.

### 3.2. Isolation

The air-dried and ground *E. antiquorum* (6 kg) was extracted exhaustively with methanol (10 L × 3) at room temperature. After evaporation of the extracts, the residue (757.2 g) was dissolved in ethanol, suspended in water then successfully partitioned with *n*-hexane, *n*-hexane-EtOAc (1:1, *v*/*v*), and EtOAc to give *n*-hexane, *n*-hexane-EtOAc (1:1, *v*/*v*), and EtOAc extracts. The EtOAc extract (180.7 g) was purified by silica gel column chromatography (CC) using *n*-hexane-EtOAc-Acetone (1:1:2, *v*/*v*/*v*) as an eluent, to afford seven major fractions, labelled A–E. Fractions B and C were investigated in our previous reports [10]. Fractions D were selected for further isolation. The fraction D (12.3 g) was loaded onto Sephadex LH-20 CC eluting with the solvent system of CH_2_Cl_2_–MeOH (1:3, *v*/*v*), yielding six fractions, D1–6. Fraction D3 (1.3 g) was applied to silica gel CC, eluted with the solvent system of *n*-hexane-EtOAc (1:1.5, *v*/*v*) to give four subfractions, D3.1–3.4. Subfraction D3.4 (227 mg) was selected for C18 reversed-phase CC using the solvent system of MeOH-H_2_O (2:1, *v*/*v*) as a mobile phase to obtain **4** (3.8 mg) and **2** (3.2 mg). Fraction D4 (1.72 g) was separated by silica gel CC using *n*-hexane-EtOAc-MeOH (2:1:0.1) to give three subfractions, D4.1–D4.3. Subfraction D4.3 (501 mg) was chromatographed by C18 reversed-phase CC with solvent H_2_O-MeOH (2:1) to afford two subfractions D4.A–D4.B. Subfraction D4.A (151.6 mg) was further purified by silica gel CC using *n*-hexane-EtOAc-Acetone (4:3:2, *v*/*v*/*v*) as an eluent and **3** (3.5 mg) were obtained. Subfraction D4.B (249.7 mg) was further purified using the same manner to afford **1** (4.0 mg) and **6** (3.3 mg). Fraction D5 (2.1 g) was applied to silica gel CC with solvent system of *n*-hexane-EtOAc-Acetone (1:1.5:1, *v*/*v*/*v*) as a mobile phase to yield five subfractions, D5.1–D5.5. Subfraction D5.3 (313.2 mg) was selected for reversed-phase CC, eluted with the solvent system of MeOH-H_2_O (2:1, *v*/*v*) to give **5** (3.9 mg) and **7** (4.1 mg).

#### 3.2.1. *Ent*-1*α*,3*α*,16*β*,17-tetrahydroxyatisane (**1**)

Colorless gum. [*α*]^20^_D_ + 117 (c 0.1, MeOH). HR-ESI-MS m/z 301.2336 (calcd. for C_20_H_34_O_4_Na, 301.2355); ^1^H-NMR (CD_3_OD, 400 MHz) δ_H_ 3.52 (1H, t, *J* = 3.2, H-1), 3.48 (1H, d, *J =* 11.6, H-17a), 3.41 (1H, t, *J =* 3.2, H-3), 3.34 (1H, d, *J* = 11.2, H-17b), 2.24 (1H, dt, *J* = 15.1, 2.8, H-2a), 2.07 (1H, m, H-9), 2.04 (1H, m, H-13a), 1.89 (1H, m, H-14a), 1.85 (1H, m, H-2b), 1.81 (1H, m, H-12), 1.67 (1H, m, H-11a), 1.60 (1H, m, H-5), 1.49 (1H, m, H-11b), 1.47 (2H, m, H-6), 1.43 (1H, m, H-7a), 1.18 (1H, m, H-13b), 1.15 (1H, m, H-7b), 1.12 (2H, s, H- 15), 1.00 (3H, s, H-20), 0.97 (3H, s, H-19), 0.85 (3H, s, H-18), 0.78 (1H, m, H-14b).^13^C-NMR (CD_3_OD, 100 MHz) δ_C_ 78.8 (C-3), 75.2 (C-16), 73.6 (C-1), 69.7 (C-17), 54.1 (C-15), 44.5 (C-5), 44.0 (C-9), 42.8 (C-10), 40.5 (C-7), 38.8 (C-4), 34.0 (C-8), 33.1 (C-12), 30.3 (C-2), 29.1 (C-19), 28.6 (C-14), 24.5 (C-11), 23.3 (C-13), 22.3 (C-18), 19.4 (C-6), 15.1 (C-20).

#### 3.2.2. Ethyl *ent-*3,4-*seco*-4,16*β*,17-trihydroxyatisane-3-carboxylate (**2**)

Colorless gum. [*α*]^20^_D_ + 121 (c 0.1, MeOH). HR-ESI-MS *m*/*z* 405.2628 (calcd. for C_22_H_38_O_5_Na, 405.2617); ^1^H-NMR (CD_3_OD, 400 MHz) δ_H_ 4.09 (2H, q, *J* = 7.2 Hz, H-21), 3.49 (1H, d, *J* = 11.2 Hz, H-17a), 3.35 (1H, d, *J* = 11.6 Hz, H-17b), 2.55 (1H, m, H-2a), 2.28 (1H, m, H-2b), 2.18 (1H, m, H-1a), 1.95 (1H, m, H-11a), 1.90 (1H, m, H-14a), 1.84 (1H, m, H-12), 1.65 (2H, m, H-13), 1.56 (1H, m, H-1b), 1.51 (2H, m, H-6), 1.47 (1H, m, H-9), 1.39 (1H, m, H-5), 1.33 (1H, m, H-7a), 1.27 (3H, s, H-18), 1.27 (3H, s, H-19), 1.25 (3H, t, *J* = 7.2 Hz, H-22), 1.21 (1H, m, H-11b), 1.18 (1H, m, H-15a), 1.14 (3H, s, H-20), 1.11 (1H, m, H-7b), 1.09 (1H, m, H-15b), 0.84 (1H, m, H-14b). ^13^C-NMR (CD_3_OD, 100 MHz) δ_C_ 177.0 (C-3), 76.1 (C-4), 75.0 (C-16), 69.7 (C-17), 61.4 (C-21), 53.5 (C-5), 53.3 (C-15), 45.1 (C-9), 42.3 (C-10), 40.0 (C-7), 35.5 (C-12), 35.1 (C-1), 34.1 (C-8), 32.7 (C-19), 30.1 (C-2), 28.4 (C-18), 27.9 (C-14), 24.2 (C-13), 23.9 (C-11), 23.4 (C-6), 19.0 (C-20), 14.6 (C-22).

### 3.3. Alpha-Glucosidase Inhibition Assay

*Saccharomyces cerevisiae* α-glucosidase (E.C 3.2.1.20), acarbose, and 4-nitrophenyl *β*-D-glucopyranoside (*p*NPG) were obtained from Sigma-Aldrich Co (Saint Louis, MI, USA). The alpha-glucosidase (0.2 U/mL) and substrate (5.0 mM *p*NPG) were dissolved in 100 mM pH 6.9 sodium phosphate buffer [17]. The inhibitor (50 µL) was preincubated with alpha-glucosidase at 37 °C for 20 min, and then the substrate (40 µL) was added to the reaction mixture. The enzymatic reaction was carried out at 37 °C for 20 min and stopped by adding 0.2 M Na_2_CO_3_ (130 μL). Enzymatic activity was quantified by measuring absorbance at 405 nm (CLARIOstar plus, BMG LABTECH, Ortenberg, Germany). All samples were analyzed in triplicate at five different concentrations around the IC_50_ values, and the mean values were retained. The inhibition percentage (%) was calculated by the following equation:Inhibition (%) = [1 − (A_sample_/A_control_)] × 100.(1)

### 3.4. Inhibitory Type Assay of ***3*** on Alpha-Glucosidase

The mechanisms of inhibition of alpha-glucosidase by **3** were determined by Lineweaver–Burk plots (Microsoft Excel 2010, Redmond, WA, USA), using methods similar to those reported in the literature. Enzyme inhibition due to various concentrations of the **3** were evaluated by monitoring the effects of different concentrations of the substrate. For Lineweaver–Burk double reciprocal plots 1/enzyme velocity (1/V) vs. 1/substrate concentration (1/[S]), the inhibition type was determined using various concentrations of *p*NPG (1 mM, 2 mM, and 4 mM) as a substrate in the presence of different concentrations of the test compound (0, 13.8, 27.7, and 55.5 µM). The experiments were carried out in three replicates. The mixtures were incubated at 37 °C and the optical density was measured at 405 nm every 1 min for 30 min with the Clariostar Labtech microplate reader (Ortenberg, Germany). Optimal concentrations of the tested compound were chosen based on the IC_50_ value. The inhibition constants were obtained graphically from secondary plots (Microsoft Excel 2010, Redmond, WA, USA).

### 3.5. Isolation and Structure Elucidation of the Compounds

Gravity column chromatography was performed on silica gel 60 (0.040–0.063 mm, Merck, Darmstadt, Germany). Thin-layer chromatography (TLC) for checking chromatographic patterns of fractions and isolated compounds was carried out on silica gel 60 F_254_ (Merck, Darmstadt Germany) and spots were visualized by spraying with 10% H_2_SO_4_ solution followed by heating. Specific rotations were obtained on a Jasco P-1010 polarimeter (Oklahoma City, OK, USA). The HRESIMS were recorded on a MicroOTOF-Q mass spectrometer (Bruker, MA, USA). The NMR spectra were measured on a Bruker Avance 500 MHz spectrometer (Bruker, MA, USA).

## 4. Conclusions

From the Vietnamese plant *E. antiquorum*, seven alpha-glucosidase inhibitors were isolated and elucidated, including *ent*-1*α*,3*α*,16*β*,17-tetrahydroxyatisane (**1**), ethyl *ent*-3,4-*seco*-4,16*β*,17-trihydroxyatisane-3-carboxylate (**2**), *ent*-atisane-3-oxo-16*β*,17-acetonide (**3**), *ent*-3*α*-acetoxy-16*β*,17-dihydroxyatisane (**4**), *ent-*16*β*,17-dihydroxyatisane-3-one (**5**), calliterpenone (**6**), and ingol 12-acetate (**7**). To the best of our knowledge, compounds **1**–**6** were isolated from this species for the first time. Compounds **1** and **2** were new compounds. Compound **3** exhibited the highest inhibitory activity against yeast alpha-glucosidase inhibitory activity with IC_50_ value of 69.62 µM. The kinetic mechanism of **3** indicated that it retarded alpha-glucosidase in a noncompetitive manner. In this study, compound **3** showed the most powerful yeast α-glucosidase inhibitory activity. However, it could not be considered a potential antidiabetic until other studies were performed.

## Figures and Tables

**Figure 1 molecules-26-02257-f001:**
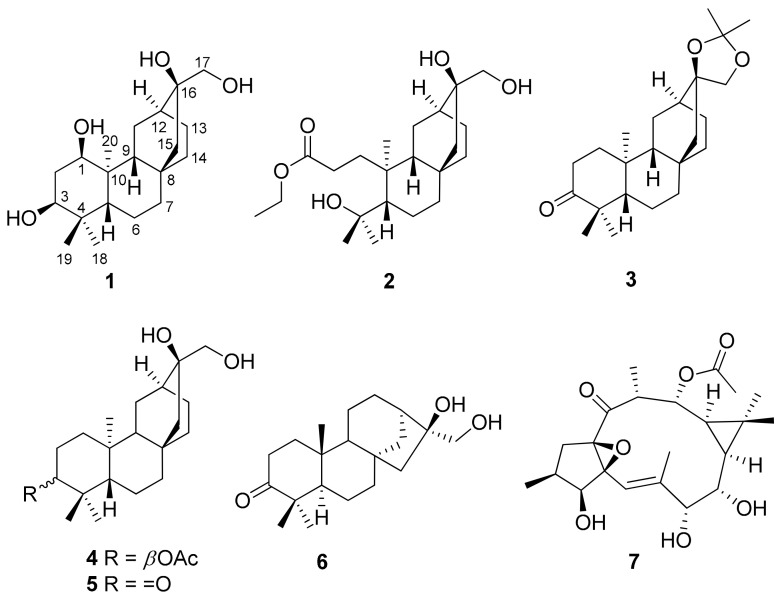
Chemical structures of **1**–**7**.

**Figure 2 molecules-26-02257-f002:**
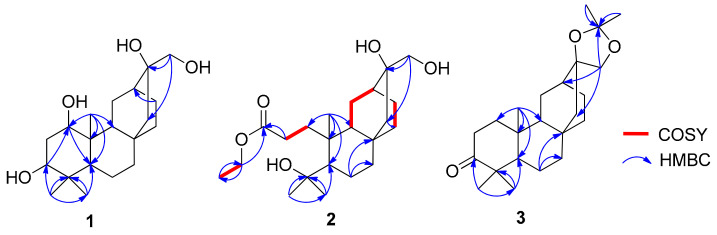
Key COSY and HMBC correlations of **1**–**3**.

**Figure 3 molecules-26-02257-f003:**
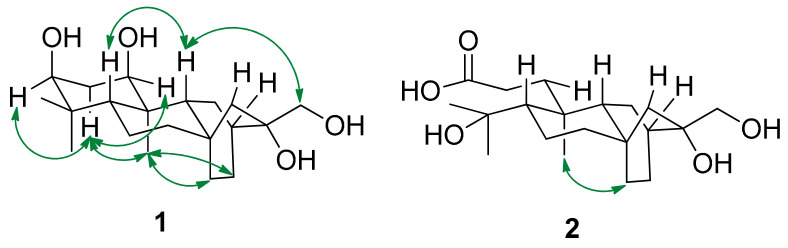
Key NOESY correlations of **1** and **2**.

**Figure 4 molecules-26-02257-f004:**
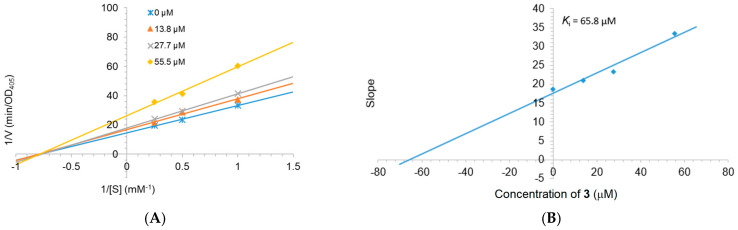
Lineweaver–Burk plot for alpha-glucosidase inhibition by **3** (**A**) and the secondary plot of slope vs. the inhibitor concentration (**B**).

**Table 1 molecules-26-02257-t001:** Alpha-glucosidase inhibitory activity of **1–4**, **6**, and **7**.

Compound	IC_50_ (µM)
**1**	125.20
**2**	130.80
**3**	69.62
**4**	102.18
**6**	156.14
**7**	115.23
Acarbose	332.5

## Data Availability

All data supporting this study is available in the manuscript and the Appendix A.

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
