# Peer review of "Alpha-Glucosidase Inhibitory Diterpenes from Euphorbia antiquorum Growing in Vietnam"

_molecules, 2021, doi:10.3390/molecules26082257_

Round 1
Reviewer 1 Report
The paper reports on the isolation and identification of seven diterpenes derived from euphorbia antiqorum, which possess potent alpha-glucosidase inhibitor activity and are interesting for further investigation. The research findings are very clear and well presented.
Some minor corrections:
Lines 65 and 66, if possible, please indicate whether the coupling constants are geminal or vicinal e.g. 2JH-H or 3JH-H.
Line 99: Hyphen between IC50 and 69.62 µM
Line 117: For better understanding please write the full name of pNPG and put the abbreviation in brackets.
Line 208: The positive control (acarbose) has to be mentioned: "... IC50 value of 69.6 µM using (?) as the positive control."
Author Response
Responding to Editor and Reviewer¢s comments
Journal: Molecules (ISSN 1420-3049)
Manuscript ID: Molecules-1181719
ALPHA-GLUCOSIDASE INHIBITORY DITERPENES FROM EUPHORBIA ANTIQUORUM GROWING IN VIETNAM
We thank for the corrections of the Reviewers and have tried to adjust the manuscript.
All errors corrected by the Reviewer were adjusted and highlighted in yellow color in the manuscript.
Reviewer: 1
The paper reports on the isolation and identification of seven diterpenes derived from euphorbia antiqorum, which possess potent alpha-glucosidase inhibitor activity and are interesting for further investigation. The research findings are very clear and well presented.
The authors highly appreciated the Reviewer for this positive comment.
Some minor corrections:
Lines 65 and 66, if possible, please indicate whether the coupling constants are geminal or vicinal e.g. 2JH-H or 3JH-H.
Line 99: Hyphen between IC50 and 69.62 µM
Line 117: For better understanding please write the full name of pNPG and put the abbreviation in brackets.
Line 208: The positive control (acarbose) has to be mentioned: "... IC50 value of 69.6 µM using (?) as the positive control."
The authors are indebted for these valuable remarks. We revised following the suggestion.

Reviewer 2 Report
The elucidation of the structures appears sound. However, to claim the compounds as inhibitors you must give the source of the alpha-glucosidase as acarbose used as the positive control does not strongly inhibit all glucosidases. The glucosidase inhibition being better than acarbose as described is therefore meaningless. You must give the enzyme source as well as have at least one other glycosidase for comparison; the compounds may inhibit almost any enzyme by strong binding? Such binding would mean the compounds will not significantly reach alpha-glucosidases involved in blood sugar control.
The importance and merit of the manuscript will increase greatly if you can improve the reporting of the glucosidase data and add other enzymes.
Author Response
Reviewer: 2
The elucidation of the structures appears sound.
The authors highly appreciated the Reviewer for this positive comment.
However, to claim the compounds as inhibitors you must give the source of the alpha-glucosidase as acarbose used as the positive control does not strongly inhibit all glucosidases. The glucosidase inhibition being better than acarbose as described is therefore meaningless. You must give the enzyme source as well as have at least one other glycosidase for comparison; the compounds may inhibit almost any enzyme by strong binding? Such binding would mean the compounds will not significantly reach alpha-glucosidases involved in blood sugar control.
The importance and merit of the manuscript will increase greatly if you can improve the reporting of the glucosidase data and add other enzymes.
The authors are grateful to the reviewer for the precious remark. We added the enzyme source of alpha-glucosidase. As regards to other suggested tests, the isolated compounds, especially compound 3 have minute amounts, thus we do not have enough the pure compounds for other tests. The authors sincerely apologized for this.
Round 2
Reviewer 2 Report
I sympathise over the small amounts of compounds for assays using another glycosidase. However, it means the conclusion that the molecules might have anti-diabetic potential cannot be made without this. It is clear that acarbose does not inhibit yeast alpha-glucosidase and so it not valid to say your compounds are better inhibitors. I have noted there are other published papers doing the same mistake orshowing lack of knowledge. If an author wants to compare with acarbose they need to use amylase or an intestinal disaccharidase.
I do think the compounds are of interest as structures. It is perhaps worth mentioning the alpha-glucosidase activity but not saying this means they are better than acarbose. You should also mention that other glycosidases were not tried to show selectivity because not enough compound was available.
I attach my comments on the first version of the manuscript as there were some minor changes noted. You may have already amended these in the new version.

Author Response
I sympathise over the small amounts of compounds for assays using another glycosidase. However, it means the conclusion that the molecules might have anti-diabetic potential cannot be made without this. It is clear that acarbose does not inhibit yeast alpha-glucosidase and so it not valid to say your compounds are better inhibitors. I have noted there are other published papers doing the same mistake orshowing lack of knowledge. If an author wants to compare with acarbose they need to use amylase or an intestinal disaccharidase. I do think the compounds are of interest as structures. It is perhaps worth mentioning the alpha-glucosidase activity but not saying this means they are better than acarbose. You should also mention that other glycosidases were not tried to show selectivity because not enough compound was available.
The authors highly appreciated the Reviewer for the valuable comment. The authors fully agreed with the Reviewer. We revised following the suggestion.
I attach my comments on the first version of the manuscript as there were some minor changes noted. You may have already amended these in the new version.
The authors thank the reviewer for the remarks. They were done.
